# The Human Microbiome and Its Role in Musculoskeletal Disorders

**DOI:** 10.3390/genes14101937

**Published:** 2023-10-14

**Authors:** Khaled Aboushaala, Arnold Y. L. Wong, Juan Nicolas Barajas, Perry Lim, Lena Al-Harthi, Ana Chee, Christopher B. Forsyth, Chun-do Oh, Sheila J. Toro, Frances M. K. Williams, Howard S. An, Dino Samartzis

**Affiliations:** 1Department of Orthopedic Surgery, Rush University Medical Center, Chicago, IL 60612, USA; khaled_a_aboushaala@rush.edu (K.A.); jnbarajas04@gmail.com (J.N.B.); perryllim@gmail.com (P.L.); ana_chee@rush.edu (A.C.); chundo_oh@rush.edu (C.-d.O.); sheila_j_toro@rush.edu (S.J.T.); howard.an@rushortho.com (H.S.A.); 2International Spine Research and Innovation Initiative, Rush University Medical Center, Chicago, IL 60612, USA; 3Department of Rehabilitation Sciences, The Hong Kong Polytechnic University, Hong Kong SAR, China; arnold.wong@polyu.edu.hk; 4Department of Microbial Pathogens and Immunity, Rush University Medical Center, Chicago, IL 60612, USA; lena_al-harthi@rush.edu; 5Department of Internal Medicine, Rush University Medical Center, Chicago, IL 60612, USA; christopher_forsyth@rush.edu; 6Department of Twin Research, King’s College London, London WC2R 2LS, UK; frances.williams@kcl.ac.uk

**Keywords:** microbiome, musculoskeletal, gastrointestinal, therapeutics, immune system, disc degeneration, pain

## Abstract

Musculoskeletal diseases (MSDs) are characterized as injuries and illnesses that affect the musculoskeletal system. MSDs affect every population worldwide and are associated with substantial global burden. Variations in the makeup of the gut microbiota may be related to chronic MSDs. There is growing interest in exploring potential connections between chronic MSDs and variations in the composition of gut microbiota. The human microbiota is a complex community consisting of viruses, archaea, bacteria, and eukaryotes, both inside and outside of the human body. These microorganisms play crucial roles in influencing human physiology, impacting metabolic and immunological systems in health and disease. Different body areas host specific types of microorganisms, with facultative anaerobes dominating the gastrointestinal tract (able to thrive with or without oxygen), while strict aerobes prevail in the nasal cavity, respiratory tract, and skin surfaces (requiring oxygen for development). Together with the immune system, these bacteria have coevolved throughout time, forming complex biological relationships. Changes in the microbial ecology of the gut may have a big impact on health and can help illnesses develop. These changes are frequently impacted by lifestyle choices and underlying medical disorders. The potential for safety, expenses, and efficacy of microbiota-based medicines, even with occasional delivery, has attracted interest. They are, therefore, a desirable candidate for treating MSDs that are chronic and that may have variable progression patterns. As such, the following is a narrative review to address the role of the human microbiome as it relates to MSDs.

## 1. Introduction

A collection of microorganisms, such as bacteria and fungi, that live on the surface of the host body or within is known as the human microbiota [1]. Microbial cells that inhabit the human body, including the mucosal and skin environments, outnumber our somatic cells and carry significantly more genes than our human genome [2]. The human body contains 500–1000 bacterial species at any given moment, although the number of unique genotypes (subspecies) may be larger in magnitude [3]. Millions of genes make up each bacterial strain’s genome, giving them much greater genetic variety and flexibility than the human genome. There is a large inter-individual variability in the composition of the microbiota. People can have vastly different collections of microorganisms, yet very little is known as to what governs such variability. Importantly, although it remains uncertain how intra- or inter-individual microbial diversity affects an individual’s well-being, health preservation, or disease inception and progression, changes in the microbiome and how it interacts with the gastrointestinal, nervous, immune, and endocrine systems are well recognized and are associated with a number of illnesses, including major depressive disorder, cancer, and inflammatory bowel disease [4,5].

The study of the human microbiome has now reached a key juncture. In the United States, the prevalence of many noncommunicable or chronic diseases has risen in recent decades, owing to longer lifespans and altered lifestyle (e.g., diet, activity levels, etc.). Changes in diet (e.g., fiber content, processed foods, etc.), sanitation (e.g., chlorinated water and filtered), and antibiotic use have significantly altered the gut’s microbial population in the United States during the last 100 years [6]. These findings have led to the premise that changes in the gut microbiota, secondary to lifestyle changes, are a significant factor associated with variations in the prevalence of different diseases over time [7,8,9]. Recent advances in sequencing technology have facilitated the conduction of systematic and comprehensive research on the human microbiome [10]. DNA-based taxonomy allows for identification of microbial communities at species level, whereas microbial metatranscriptomics and metagenomics can investigate the actively transcribed microbial genes and microbial genomes, respectively [11,12,13,14]. Direct and indirect crosstalk between the gut microbiota and the host immune system is now better understood as an adaptive evolutionary process between both microbial communities and mammalian species [15,16]. 

The efficacy of lifestyle treatments aimed at reducing the effects of aging on the musculoskeletal system or the onset of illness can be significantly influenced by the gut microbiota and their metabolites. Studies have noted that gut microbiota are associated and have implications with various musculoskeletal disorders (MSDs), such as osteoporosis [17,18,19,20], sarcopenia [21], osteoarthritis [22], and rheumatoid arthritis [15] (Table 1 and Table 2). MSDs are recognized as one of the most common workplace health concerns, significantly impacting working professionals’ quality of life [23]. 

Currently, little is known about the interplay between the human microbiome and MSDs. The current narrative review evaluates emerging data on the role of the human microbiome in relation to MSDs. To our knowledge, a singular and comprehensive review on the role of the microbiome upon MSDs has not, to date, been addressed. Risk factors between many musculoskeletal conditions tend to overlap, suggesting crosstalk of these mechanisms. This crosstalk provides cause and reason to believe that a review of the microbiome in relationship to MSDs in this depth and coverage is needed and would provide a starting point for discussion and future research. This review is targeted to clinicians (e.g., orthopaedists, rheumatologists, physiatrists, physical therapists, etc.) and basic science researchers interested in the topic of microbiome and MSDs.

## 2. Human Microbiome

Microbiota is a term used to describe the microorganisms that live in a certain environment [38], while the “microbiome” is a collection of genomes from those microorganisms. The human microbiota is composed of approximately 10 to 100 trillion symbiotic microbial cells residing within each person, predominantly bacteria in the gut. The human microbiome is made up of the genes that comprise the genetic material of such microbial cells [39]. Multiple microbiome projects have been developed globally to better understand the roles of these symbiotic microorganisms and their effects on human health [40,41]. The study of the human microbiome has now reached a key juncture. The field is moving away from descriptive analyses and towards mechanistic and interventional studies [42]. These developments have also sparked a boom in translational research and significant expenditures in both academic and corporate research, including that of so-called “big pharma”. A revolution in customized medicine has sparked this organized endeavour in clinical microbiome research. Cancer genome sequencing is one instance of customized medicine since it enables quick identification of a patient’s unique treatment plan that will provide beneficial results. Our ability to define the microbiome quickly and consistently may provide a platform for the development of new diagnostic biomarkers and non-invasive treatments as well as refined predictive outcome modeling of patient care [43].

## 3. Microbiome in Health and Disease

Since the inception of the study of the human microbiome back in the 17th century, its potential significance in health and disease has gained attention in the scientific community, yet there is still so much to explore. The current cutting-edge sequencing methods allow for the investigation of microbiota composition and its potential impact on the human body. Over 100 trillion prokaryotic cells have been found to support human biological activities. Each individual is carrying 3 to 6 pounds of bacteria and other microbes at all times, with approximately 3 million protein-coding genes [44]. There are various microbial niches that are in harmony with the normal physiologic processes of the human body [45]. The host’s homeostasis is significantly influenced by the gut microbiota. [46]. Human physiology, fat storage, immune system development, behavior, digestion, angiogenesis regulation, detoxification, and development processes all depend on the gut microbiome [47,48]. Enzymes essential for the breakdown of otherwise indigestible food components and the production of vitamins are encoded by some microbes in the gut microbiome [49,50]. Actinobacteria, Firmicutes and Bacteroidetes are the most prevalent microbiota phyla in a healthy gut, while Verrucomicrobia, Proteobacteria, Fusobacteria, and Cyanobacteria are frequently absent [41,51]. There have also been reports of Eucarya (mostly yeasts), methanogenic archaea (particularly *Methanobrevibacter smithii*) and numerous phages in a healthful gut microbiome [52]. 

Multiple factors can affect the microbiota. While the microbiota content varies greatly from one individual to the next [41], the microbiota is a highly dynamic ecosystem that can be altered by a variety of circumstances, such as pharmaceutical treatments, illness [53], age, travel [54], hormonal cycles [55] and diet [56]. Furthermore, according to He et al. [57] microbial patterns vary across geographical regions. Within the same geographical region, ethnicity plays the largest role in explaining inter-individual differences in microbiome composition [58]. The gut microbiota can have a major impact on health and become disrupted during certain pathological conditions. Environmental factors can also disrupt gut microbial communities in genetically vulnerable individuals, resulting in dysregulation of the host’s innate and adaptive immune systems, resulting in the emergence of different illnesses [45]. The interplay between the immune system and the altered gut microbiota has been shown to affect the etiology of various diseases in recent years, including cancer, inflammatory bowel syndrome, metabolic syndrome and nonalcoholic fatty liver disease [59,60,61,62,63] and many autoimmune diseases (e.g., rheumatoid arthritis, multiple sclerosis, autoimmune hepatitis, spondyloarthritis and type 1 diabetes) [64,65,66,67,68,69]. The gut microbiota is highly organized, and it has a wide range of protective, structural, metabolic, and immunological effects both within the gut and across the body [70]. Metabolites contributed by gut microbiota and their respective functions are illustrated in Table 3. 

Many gut microbiota metabolite end-products have critical roles in immunological processes, host metabolic balance and neurology, highlighting the extended genome’s basic involvement in human disease [71]. The most compelling representations of the role of gut bacterial metabolites in cardiometabolic illnesses come from comprehensive studies of choline and methylamine metabolism in humans and animal models [72]. Bile acids are steroid molecules that are generated in the liver from cholesterol and then processed by the gut microbiota into secondary bile acids [71].

**Table 3 genes-14-01937-t003:** Metabolites by the gut microbiota and their respective functions. Adapted from Kho and Lal [73].

Metabolites	Functions
Polyamines, e.g., putrescine, spermidine, and spermine [74,75,76,77,78]	Maintain a high rate of intestinal epithelial cell proliferation. Dysregulated polyamine metabolism may promote the growth of cancer. Promote the production of intercellular junction proteins (occludin, zonula occludens-1 (ZO-1), and E-cadherin) to improve intestinal barrier integrity and function. Spermine suppresses pro-inflammatory M1 macrophage activation and promotes intestinal and systemic adaptive immune system development.
Vitamins, e.g., thiamine (B1), riboflavin (B2), pantothenic acid (B5), niacin (B3), pyridoxine (B6), folate (B11–B9) biotin (B7), cobalamin (B12), and menaquinone (K2) [79,80]	The creation of red blood cells, the generation of energy, and the role of cofactors in many metabolic processes. DNA methylation, repair, and replication, which controls cell division. The creation of vitamins, amino acids, and nucleotides. Enhance immune functioning.
Phenolic derivatives, e.g., 4-OH phenyl acetic acid, equol, urolithins, enterolactone, enterodiol, 8-prenylnaringenin, 2-(3,4-dihydroxyphenyl) acetic acid, 3-(4-hydroxyphenyl) propionic acid, and 5-(3,4-dihydroxyphenyl) valeric acid [81,82,83]	Antimicrobial effects: suppressing harmful bacteria, affecting the makeup of the gut microbiota, and maintaining intestinal health. Protective effect against oxidative stress. Estrogen-modulating effect. λ platelet aggregation inhibition effect. The anti-inflammatory and cancer-prevention properties of urolithin are present.
Choline metabolites, e.g., betaine and choline, and trimethylamine N-oxide (TMAO) [84,85]	Alter glucose homeostasis and lipid metabolism. Cause cardiovascular disease and non-alcoholic fatty liver disease.
Bile acid metabolites, e.g., lithocholic acid (LCA) and deoxycholic acid (DCA) [86]	Exhibit antimicrobial activities. Activate host nuclear receptors and cell signaling pathways; regulate bile acid, cholesterol, glucose, lipid, and energy metabolism.
Indole derivatives, e.g., indole, indoxyl sulfate, and indole-3-propionic acid (IPA) [87,88,89]	IPA functions as a potent antioxidant, a blocker of amyloid-beta fibril production, and a cytoprotective agent against a range of oxidants. Pregnane X receptor (PXR), a xenobiotic sensor, is how IPA controls the intestinal barrier function. Through the PXR, IPA lowers intestinal inflammation (by downregulating the pro-inflammatory cytokines TNF-) and regulates intestinal permeability and mucosal integrity (by upregulating junctional protein-coding mRNAs). The uremic toxin indoxyl sulfate, which builds up in the blood of people with compromised excretion systems.
Short-chain fatty acids (SCFAs), e.g., acetate, butyrate, propionate, hexanoate, and valerate [89,90,91]	Control host metabolic pathways by signaling from G-protein-coupled receptors GPR41 or GPR43: energy homeostasis; glucagon-like peptide 1 (GLP-1) synthesis; upregulation of leptin production. Improve insulin sensitivity and glucose tolerance. Strong histone deacetylase (HDAC) inhibitor controls the proliferation of intestinal cells. Intestinal gluconeogenesis, lipogenesis, and inhibition of fasting-induced adipose tissue in the intestinal epithelium (factor FIAF). Immunomodulatory impact, dendritic cell activation, and gut immunity.

## 4. Microbiome and Musculoskeletal Development

An aberrant or malfunctioning microbiota has been linked to a variety of illnesses and problems not related to the gut. There is growing evidence that altered gut microbiome is highly related to certain MSDs. Changes in the microbiota provide plausible potential mechanisms for generating inflammation, modifying the immune response, and affecting host metabolism, all of which may modulate the development of MSDs and frailty. Studies in this field are challenging to carry out and require thorough planning to eliminate confounding variables such host genetics, nutrition, age, and the sickness itself, all of which alter the composition of the gut microbiome [92]. When compared to adults, children have dramatically different gut flora [93]. Despite the fact that fewer bacterial species dominate the infant’s gut microbiota, a variety of variables have a role in its growth throughout the first year of life [94]. 

## 5. Gut Microbiota and Bone Health

Over the last two decades, inflammatory bowel diseases have been associated with a loss of bone mass, implying that the intestinal and skeletal systems are linked. Recently, it was shown that the microbiomes of people with osteoporosis were much more varied than those of healthy controls, particularly in terms of Firmicutes abundance [17]. In a study postmenopausal woman with osteoporosis and osteopenia were shown to have altered microbiome [30]. Furthermore, a clinical trial demonstrated that replenishing the microbiota with probiotic medication reduced bone loss in older people with low bone mineral density [95]. Conversely, a recent study showed that probiotics reduced femoral neck bone mineral density, but hip bone mineral density was unaffected [18]. 

Mice’s osteocalcin and bone strength have decreased owing to antibiotic-induced microbiota depletion, which has been related to decreased vitamin K2 levels [32]. Given the connection between the stomach and the bones, the medical world has recently become interested in probiotics, which reintroduce beneficial bacteria strains. Probiotics like the bacterial strain Lactobacillus rhamnosus GG and VSL#3, have been shown to improve bone health by repairing intestinal permeability and restoring the gut microbiota in animal research [34]. In estrogen-deficient mice, *Lactobacillus reuteri* was demonstrated to protect against bone loss when given as a supplement [96,97]. 

Disruption of gut homeostasis can induce an inflammatory immunological phenotype that affects bone metabolism. Increases in TH17 cells that release the interleukin IL-17 in the bone marrow (BM) can promote bone breakdown by facilitating the differentiation of osteoclasts (OCs) in the BM [98,99]. Yu et al. [100] examined how intestinal immune cells affected bone remodeling in hyperparathyroid mouse models. They discovered that parathyroid hormones enhanced the number of gut TH17 cells in mice with segmented filamentous bacteria in their microbiomes. These TH17 cells moved into the BM after egressing from the stomach and into circulation, causing bone degradation [100]. To demonstrate intestinal TH17 cell egress, they used an FTY720 antagonist to block the sphingosine 1 phosphate (S1P) receptor-1, which limits lymphocyte egress from the mesenteric lymph nodes, resulting in a BM TH17 cells decrease and bone loss [100]. They also found that the chemoattractant CCL20, which guides TH17 cells into the BM, was elevated in the BM, demonstrating the relevance of TH17 cell migration into the BM [100]. Although the quantity of intestinal TH17 cells was unaffected by the administration of neutralizing anti-CCL20 antibody, it prevented the decreases in bone loss and the increase in TH17 cells in the BM.

## 6. Osteoporosis

An increase in the risk of bone fracture is associated with osteoporosis, which is defined by a decline in bone strength, a measure of bone density and quality [101]. Increased inflammatory markers in plasma, such as high-sensitivity C-reactive protein (CRP), increase bone resorption, bone loss and may increased risk of fracture [102,103,104,105]. Osteoclasts, which are produced from hematopoietic stem cells, are highly specialized motile migratory bone resorptive cells. Because osteoclasts are the most important bone resorbing cells, local activation of their activity is required to prevent alveolar bone loss [106]. Mineral bioavailability from the diet is influenced by probiotics. Human studies have found that adolescents and postmenopausal women benefit from probiotics because they boost calcium absorption [107,108]. Compared to those who took placebo pills, adolescents who were given a mix of inulin-type fructans with short and long chains demonstrated considerable increase in bone density and mineral content throughout the body in the bone [109]. The flora of the distal colon ferments fructans in the intestinal tract rather than having them degraded by enzymes in the small intestine. A substance that favorably affects mineral absorption is fructans. Fructans have been found to improve the absorption of a number of minerals, including calcium and iron. [110]. Alterations in calcium absorption, according to the authors, were most likely to blame; however, changes in gut microbiota composition and immune response could also have played a role [111,112]. In adolescent girls, the administration of galacto-oligosaccharides (GOS) on calcium absorption and the intestinal microbiota was studied [112]. In a dose-dependent manner, beneficial fecal bifidobacteria numbers increased considerably. Calcium absorption increased as well, although no direct relationship with dose was observed. A different study found enhanced calcium absorption and an increase in microbiota in teenagers who consumed a soluble maize fiber diet compared to similar controls who ate normal diet; however, no changes in indicators of bone turnover were observed. These results suggest that among adolescents who eat less calcium than is advised, a moderate daily dosage of SCF, a well-tolerated prebiotic fiber, enhances short-term Ca absorption [113].

## 7. Rheumatoid Arthritis

The autoimmune disease rheumatoid arthritis (RA) affects the synovium and cartilage and is commonly accompanied with bone erosions. The prevalence of RA is rising, as is the age at which it first appears, partly due to population ageing [97,98]. Despite shorter symptomatology at presentation, older patients frequently have greater disease severity [114]. However, the reasons for this phenomenon have not been elucidated. Intestinal microbiota components [115] affect the host immunity, especially the effector T-cell development, which may influence autoimmune disease susceptibility, such as RA [116]. The gut microbiota is also known to alter with age [117]. Seropositive RA is characterized by antibodies against citrullinated proteins (ACPAs), which have been found to be harmful. According to the CIA model, Porphyromonas can hasten the course of RA in animals [118,119]. Genetic changes in the microbiota may help mitigate genetic risk [31,120]. Prevotella spp were shown to relate to the RA genotype in absence of RA, including people with high risk factor of having RA [120]. Autoantigenic citrullinated peptides secreted by oral microbiome were demonstrated to be elevated in RA patients [121]. However, in adjuvant-induced arthritis in rats, *L. casei* (ATCC334) bacteria was shown to restore bone loss indicating this to be a potential candidate for probiotics for treating RA [122].

## 8. Sarcopenia

A broad, progressive skeletal muscle condition known as sarcopenia causes loss of muscular mass and function [123]. It occurs with advancing age, and causes decreased strength, eventually leading to physical decline, incapability to carry out daily tasks, loss of independence, higher risk of falls, and an increased chance of death [124]. Sarcopenia makes recovering from bone and joint diseases more difficult, including orthopaedic surgery. Various medical conditions (e.g., cancer, malnutrition, chronic infection, and chronic heart failure) can cause muscle wasting. Inflammation and a poor nutritional state are also believed to be shared mechanisms [125]. Higher blood interleukin-6 (IL-6) and CRP levels were observed to enhance the likelihood of muscular strength loss over a 3-year period in a prospective investigation involving over 1000 participants [126].

In sarcopenic obesity, the microbiome may also play a role [127]. Increased adiposity, fat redistribution, persistent muscle fat infiltration and low-grade inflammation are all symptoms of this sarcopenia type, which has no universal diagnostic criteria [128]. The function of the gut microbiota in obesity has been studied extensively [127]. When compared to normal-weight and age-matched controls, overweight individuals have a less diverse microbiome, which could contribute to weight gain and increased inflammation in muscle [128]. Based on a study by Kang et al. [127] whereby 60 healthy and 27 sarcopenic individuals were compared, 16S rRNA sequencing data of fecal microbiota revealed reduction in microbial diversity for those with sarcopenia. More specifically, the study noted an increased abundance of Lactobacillus but a decreased abundance of Fusicantenibacter, Lachnospira, Eubacterium, Roseburia, and Lachnoclostridium in sarcopenic subjects. 

## 9. Osteoarthritis

The classic view of osteoarthritis was that it was a non-inflammatory arthropathy characterized by cartilage and bone remodeling. On the other hand, several studies have consistently demonstrated inflammation throughout the whole illness phase [129,130]. Streptococcus species within the gastrointestinal microbiome has shown to trigger endotoxin release leading to local inflammation in the knee joint with resulting pain [131]. Indeed, the synovium and chondrocytes create chemokines, cytokines and other inflammatory mediators have been detected in the synovial fluid of OA joints [129,130]. Furthermore, bacterial DNA has been found in the synovial fluid and synovial tissue of OA joints from polymerase chain reaction (PCR) tests, suggesting the potential that living bacteria are present in the joint as the illness progresses [127,132,133]. Investigations employing sequence-based microbiota assays are essential since a significant portion of gut bacteria is currently unculturable. For 12 weeks, participants in these earlier research received either 3000 mg/day of an oral glucosamine sulphate (GS) supplement or 3000 mg/day of a whole green-lipped mussel (GLM) extract. However, since these tests are digested in the colon by gut bacteria, considerable disagreement existed concerning their efficacy. The discrepancy could be due to differences in baseline microbiota between studies and individuals [22].

## 10. Intervertebral Disc Degeneration

Disc degeneration (DD), which causes low back pain, is the most disabling ailment in the world and well known potential risk factor [134,135,136,137,138,139]. Inflammation has been identified as the final common pathway leading to DD, but the mechanisms that cause it remain unknown [28,140]. Although environmental, genetic, and mechanical factors have all been implicated, subclinical infection and bacteria-induced inflammation are of particular interest [26,141,142,143,144]. Bacterial presence has been found in DD, but it has been suggested that it could be contamination rather than infection [25,29]. The existence of a unique disc biome in human intervertebral discs and disc dysbiosis in degenerative discs have been shown utilizing advanced omics technologies [27,145]. A comparative analysis of gut microbiota in 36 obese or overweight individuals with or without back pain have revealed direct correlation with altered gut microbiota and back pain, which may be attributed to increased inflammation [146]. More specifically, genera Adlercreutzia, Roseburia, Uncl. Christensenellaceae were observed abundantly in obese or overweight individuals with back pain. However, treating a mouse model for lumbar disc herniation with *Lactobacillus paracasei* S16 has been shown to reduce the inflammatory response [36]. Such an approach has also altered the gut microbiota and serum metabolites. Further research is required to examine the interactions between bacterial inflammation and dysbiosis and their role in DD in the setting of pain and non-painful states, potentially opening the possibility for new avenues of understanding the pathomechanisms of symptomatic degenerated discs, the development of more targeted novel treatments and more refined predictive outcomes.

## 11. Modic Changes

Modic changes (MC) are subchondral vertebral bone marrow non-neoplastic lesions observed on magnetic resonance imaging (MRI) [136,147,148,149,150]. These lesions involve both the vertebrae and the endplate of the adjacent disc and are related to DD. MC (especially MC type 1 [MC-1]) is thought to be related to low back pain (LBP) [151,152,153]. The prevalence of MC increases with age [136,154]; however, such lesions are more common among individuals with lumbar disc herniation and chronic LBP (approximately 45%) as compared to the general population (around 5% [155,156]). There are three types of MCs, which are determined based on T1- and T2-weighted MRIs. The MC-1 indicates inflammatory changes or edema in the vertebral bone marrow; MC-2 represents the replacement of normal haemopoietic bone marrow by yellow lipid marrow; and MC-3 indicates subchondral bone sclerosis and has low prevalence. Of the three types of MCs, MC-1 is associated with a higher prevalence of chronic LBP [157,158]. Although multiple factors (e.g., biomechanical changes or degeneration) have been proposed to be related to the development of MCs, some animal and human studies indicated that MCs might be attributed to low-virulent intra-discal infection [141,142,159,160,161]. For instance, Chen et al. [162] showed that the inoculation of human-derived *Propionibacterium acnes* into the L5, L6, and L7 intervertebral discs of rabbits caused gradual disc degeneration, adjacent endplate disruption, and MC. Although the mechanisms underlying *P. acnes* inducing MC are uncertain, it has been hypothesized that *P. acnes* secreted propionic acid may dissolve fatty bone marrow and may result in MC. It has also been hypothesized that the presence of *P. acnes* induces monocytes to create pro-inflammatory cytokines (such TNF- and IL-1) [162]. that are responsible for edema of subchondral bone marrow and potential damage of the endplates [163]. 

Human randomized controlled trials were conducted to evaluate the use of 90 or 100 days of Amoxicillin-clavulanate or Amoxicillin alone in improving the symptoms or MC in patients with chronic LBP and concomitant MC [162,164,165,166]. They found that antibiotics were significantly more effective than placebo in improving pain or disability immediately after the treatment or at the one-year follow-up in patients with MC-1. However, there were inconsistent findings regarding the clinical significance of these findings. Nonetheless, these studies underscore that perhaps bacterial infection may be related to spine health and that more targeted, precision-based and imaging approaches with respect to various structural spine phenotypes are needed. 

## 12. Scoliosis

Scoliosis is the most common rotational malformation of the spine, affecting about 1 to 4 percent of adolescents worldwide [167]. Cardiovascular risk, respiratory failure and death are associated with severe scoliosis [168]. The etiology of scoliosis remains uncertain but is believed to be multifactorial [167]. For example, central nervous system difficulties, genetic factors, skeletal spinal growth, bone metabolism abnormalities, metabolic pathways, biomechanics, proprioceptive deficits, and other factors have all been postulated as possible explanations for its pathophysiology [169,170]. Clinical symptoms and pathological abnormalities of scoliosis suggest that metabolic dysfunction and biochemical variables may be involved in its development [171,172,173]. Recent research reveals that the gut microbiota plays a significant role in regulating metabolic and biochemical processes that support bone production and skeletal development [37,174]. Recent observations in the distinct gut microbiota profiles in patients with scoliosis as compared to non-scoliotic controls have led to the development of a novel hypothesis [175]. Specifically, the changes in plasma proteins and the amount of fecal Prevotella were found to be positively associated with the Cobb angles of patients with scoliosis [175]. However, there is insufficient evidence to substantiate the direct involvement of gut microbiome on the onset or progression of scoliosis. Future research should investigate the gut microbiota profiles of patients with rapid progressive scoliosis to improve our understanding regarding the pathophysiology or progression of scoliosis.

## 13. Microbiome and Inflammatory Conditions

### 13.1. Septic Arthritis

Most commonly, *Neisseria gonorrhoeae* and *Staphylococcus aureus* (*S. aureus*) cause septic arthritis in individuals of all ages [176,177,178]. The most common cause of bloodstream infections globally is *S. aureus* [179]. However, due to a continuous extension from a nearby soft tissue infection, *S. aureus* can also result in septic arthritis. After minor trauma or in medical settings when a central venous catheter or peripheral intravenous device is used, bloodstream infections brought on by *S. aureus* are frequent [180]. Numerous virulence traits of *S. aureus* enable it to penetrate the joint space and evade human defenses, leading to symptomatic infection. These include protein A, collagen-binding protein, clumping factors A and B, and bone sialoprotein-binding protein, which all have similar mechanisms of action involving extracellular matrix adhesion and cause osteomyelitis and septic arthritis [181]. *S. aureus*, however, is regarded as a human commensal since up to 30% of asymptomatic individuals have it in their noses. Methicillin-resistant *S. aureus* (MRSA) septic arthritis is a prominent subgroup of *S. aureus* septic arthritis [182]. According to a systematic analysis, children under the age of two and African Americans are more commonly affected by CA-MRSA bone and joint infections [183]. Although the causes are unknown, they could have something to do with access to care or unusual presentations of bone and joint infections in very young infants [174]. A patient with septic arthritis of the hip or knee who worked in a medical facility was more likely to get MRSA than methicillin-sensitive *S. aureus*, according to studies [184].

### 13.2. Osteomyelitis

Osteomyelitis is a type of bone inflammation and infection. When a bacterial or fungal infection enters bone through the circulation or surrounding tissue, it is known as osteomyelitis. Osteomyelitis can appear suddenly or gradually [185]. In the United States, it is estimated that 2 to 5 per 10,000 persons have osteomyelitis regardless of age [186]. Children are more likely to get long bone infections involving the arms or legs. However, older persons are more likely to develop osteomyelitis [187,188]; however, anyone at any age can get it because they have more health problems that raise the risk of infection, such as diabetes, lower immunity or orthopaedic problems that necessitate surgery. Osteomyelitis can be “hematogenous”, meaning it originated from the bloodstream to the bone, or “non-hematogenous”, meaning it is not transmitted from the bloodstream [189]. If bacteria enter the body through intravenous therapy, it is more likely to develop hematogenous osteomyelitis, especially in children. Trauma, such as a fracture or an open wound, can cause non-hematogenous osteomyelitis, which is the most common cause of non-hematogenous osteomyelitis in children. Surgical procedures (especially those involving prosthetic materials like metal pins, screws, or plates, which can harbor fungi or bacteria) may increase the risk of osteomyelitis [190]. Although the study of the role of the microbiome in the development of osteomyelitis is currently in its infancy, recent research has established early foundations as it relates to bone health of the foot, jaw and other regions related to MSDs [191,192].

### 13.3. Post-Operative Infection

Surgical site infection (SSI) and periprosthetic joint infection are major problems following orthopedic surgery. SSI have been reported to occur in 1.4 to 5.5% of cases following ankle surgery, and 0.7 to 12% of cases following spine surgery [193]. Because of the high morbidity and mortality associated with SSI, prevention is a main area of focus. Multiple risk factors, such as smoking, open fracture, obesity, alcohol abuse, and diabetes, have been identified; however, evidence suggests the carriage of *S. aureus*, a bacterial strain that is part of the normal microflora commonly found in the anterior section of the nostrils, could play a role [194]. A clinical investigation comparing over 1000 patients prophylactically treated with mupirocin to a historical control found that it might lower the infection rate in orthopedic surgery [195]. In a cohort of 18 patients who got SSI after orthopedic surgery, another clinical investigation discovered that high nasal carriage of *S. aureus* was the only substantial independent risk factor for developing *S. aureus*-related SSI [196,197]. However, their later randomized, double-blind, placebo-controlled experiment discovered that using Mupirocin nasal ointment did not shorten hospital stays or SSI caused by *S. aureus*. The gut microbiota may be involved in periprosthetic joint infection, according to animal studies. For instance, in mice, a higher percentage of those treated with oral antibiotics for disturbed gut flora developed periprosthetic joint infection [33]. The scientists hypothesized that the mice with altered gut microbiota had a suppressed immunological response to the illness. It is necessary to do further animal and clinical research to determine how the gut microbiome’s health affects the emergence of periprosthetic joint infection and SSI.

### 13.4. Discitis

Discitis is a bacterial infection of the intervertebral disc resulting from direct injection or hematogenous seeding. Identification of a causal pathogen, when possible, is crucial for guiding the therapy protocols [198,199]. Although polymicrobial infection can occur, most cases involve a single pathogen [200]. *S. aureus* is the most common isolated pyogenic pathogen [201,202,203,204]. The MRSA infection rate ranges from 10 to 40%, notably in nosocomial acquired isolates [202,205,206]. Coagulase-negative staphylococci, such as *S. epidermidis* (5 to 29%), and Streptococcus genus, have also been noted (5 to 20%) [202,205,207]. Escherichia coli (7 to 33%) is the most often grown gram-negative bacillus, which is commonly seen in immunocompromised patients [208], followed by Pseudomonas (4 to 12%) [205,207,209]. In IV drug abusers, the most common bacteria are Pseudomonas species, *S. aureus*, and *S. epidermidis*. Patients with infected endocarditis are likely to be caused by *S. viridans* and Group D streptococci.

### 13.5. Ankylosing Spondylitis

Ankylosing spondylitis (AS) is a chronic inflammatory, autoimmune condition with a complicated pathophysiology involving specific genetic markers and distinct environmental triggers, such as gut dysbiosis [24]. Metagenomic profiling of these microbiota clearly shows microbiota perturbation and AS-enriched species are shown to trigger autoimmunity due to molecular mimicry [210]. The interaction between the antigen HLA-B27 and bacterial cells, for example, can result in HLA-B27 misfolding within the endoplasmic reticulum within the bacteria leading to inactive proteins [211,212]. The production of pro-inflammatory mediators, as well as the phenomenon of autophagy, are both caused by this unfolded response [213]. Furthermore, since there is a biochemical similarity between HLA-B27-presented bacterial peptides and different self-peptides, it could lead to a cross-immune responses [211,214]. Furthermore, according to Chen et al. [215], the gut microbiota is restored in AS patients after adalimumab treatment, and this gut microbiome functions as possible biomarkers to assess the therapeutic medication.

### 13.6. Fibromyalgia and Chronic Pain

Fibromyalgia is a chronic widespread pain with both somatic and psychological symptoms [216]. Although the causes of fibromyalgia are unclear, oxidative stress, neuroinflammation, proteomics, genetics, hormonal changes, and various types of bacterial infections have been suggested to be related to fibromyalgia [217,218,219]. It is known that fibromyalgia is closely related to irritable bowel syndrome through shared genetic factors [220]. Furthermore, Marc et al. [221], revealed that people with fibromyalgia had less diversity of fecal microbes than age-matched healthy controls. A recent systematic review by Wang et al. [222] found that the composition and metabolism of gastrointestinal microbiota were associated with fibromyalgia. However, it should be noted that potential confounders may exist, such as use of drugs (e.g., non-steroidal anti-inflammatory drugs [NSAIDs]) that may affect intestinal permeability in the included studies, that may complicate the interpretation of results. Collectively, given the potential association between fibromyalgia and gut microbiota and the paucity of relevant research, future prospective studies should adopt novel technology and standardized procedures, as well as address the limitations of prior studies in order to determine the causal relation between gut health and fibromyalgia. 

Although multiple factors will affect the development or maintenance of chronic pain, it is not uncommon for patients with musculoskeletal pain to take high-dose paracetamol (>2 g per day) or NSAIDs [223]. These medications can adversely affect the gastrointestinal environment and physiology, which may lead to gut dysbiosis (i.e. bacterial imbalance). Gut dysbiosis heightens the permeability of enterocyte to bacterial endotoxins, resulting in gastrointestinal dysfunction (leaky gut) [70]. Gastrointestinal microbiota can affect the mucosal defense mechanisms (i.e., competing for mucosal colonization and metabolic substrates for the synthesis of regulatory factors) and immune system of the host. The gastrointestinal tract is constantly subjected to both internal and external challenges that can interfere the intestinal homeostasis. Toxins from adverse bacterial activity or diet, gut dysbiosis following medications or infections can disrupt the enterocyte’s barrier function, causing translocation of toxins or bacteria-derived gene products (e.g., lipopolysaccharides) across the inflamed, permeable epithelia and resulting in systemic inflammation and endotoxaemia [224]. Prolonged pro-inflammatory responses in the extra-intestinal environment will increase the risk of chronic inflammation and body pain. If this problem is not addressed timely, it may lead to chronic pain.

## 14. Potential Therapeutics Related to the Microbiome

The metabolism of several drugs often used to treat musculoskeletal disorders is mostly regulated by the microbes in the gut, which has emerged as a hub for the development of novel therapeutics. First, the gut microbiome has an impact on popular analgesic drugs, which are still a mainstay of symptomatic treatment [225]. Furthermore, because p-Cresol is a tyrosine metabolite, dietary tyrosine alteration may further increase paracetamol toxicity. Second, the bacterially produced enzyme b-glucuronidase influences the stomach and intestinal toxicity of NSAIDs. Inhibitors of this bacterial enzyme decreased the number of NSAID-induced ulcers in mice [225]. 

Since this system’s alteration might have significant effects, especially in older people who are more susceptible to gastrointestinal bleeding, human trials proving this adjustment are urgently needed. For example, evidence from clinical trials investigating the use of epimedium-derived phytoestrogen flavonoids (EPFs) to support bone health noted a significant decrease in bone mineral density loss in late postmenopausal females in comparison to placebo participants [226]. The trial’s authors made a concentrated effort to standardize the diets of the volunteers, which may have lessened variation in the patients’ microbiota. Rats’ gut microorganisms substantially metabolize epimedii, hence the effectiveness may vary depending on the host microbiota [227]. A more recent Chinese multicenter trial indicated a significant effect after just 6 months, however the diet was not consistent throughout the study [228]. More study is required to determine if this medication is effective when used with a Western host microbiome and diet. Though this will only have an impact on therapy if absorption is postponed until the colon, in vitro models have demonstrated that some drugs used to treat bone and joint problems are digested by the gut flora. Some drugs, such the osteoporosis treatment strontium, are affected by food, especially alginates [229], which are subsequently digested by specialized bacteria, resulting in a complex interaction between the host-specific diet, medication, and microbiota profile. The severe bisphosphonate side effect of osteonecrosis of the jaw has been linked to changes in the oral microbiota in early epidemiological studies, but new data suggests that the connection is not causative but rather reflects changes in systemic immunity [230].

## 15. Future Directions

The next critical stages in comprehending the function of the microbiome are preclinical investigations that more precisely pinpoint the mechanisms linking the microbiome to the health of musculoskeletal tissues. Future clinical studies should examine the potential efficacy of microbiome-based therapies for various MSDs, and measure gut microbiota in these patient populations. Once a more complete understanding of the mechanisms relating the gut microbiome to musculoskeletal tissues is attained, appropriate microbiome-based interventions, such as fecal microbiota transplantation or and prebiotics, can be determined for various patients. In addition, with the evolving understanding of deep analytics and artificial intelligence solutions as well as integrative and multi-omics approaches, a deeper understanding of an individual’s microbiome profile may be possible and lend to more personalized patient care approaches. In fact, such platforms can further contribute to novel preventative measures, unique diagnostics, and informative predictive modeling as they pertain to MSD. 

## 16. Conclusions

The gut microbiome is now widely acknowledged in the scientific community as a contributor to the onset of illness and a prospective target for therapeutic treatments and prognostic prediction. Despite being well acknowledged in other medical fields, the gut microbiome has only just begun to be recognized as a factor in musculoskeletal health. With the explosion of deep phenotyping and analytics, and the understanding of the integration of various omics approaches, the microbiome is an emerging research area that may lead to a paradigm shift in the spectrum of personalized care for patients with different MSDs. As such, this will ultimately improve patient care, decrease healthcare costs and lead to a more productive society. 

## Figures and Tables

**Table 1 genes-14-01937-t001:** Microbiome and musculoskeletal disorders in human studies.

Author	Population Size	Population Characteristics	Males:Females	Microbe	Musculoskeletal Disorder	Findings
Cardoneanu et al. (2021) [24]	60	32 healthy controls, 28 ankylosing spondylitis	29:31	Several microbes	Ankylosing Spondylitis	Decreased intestinal bacterial diversity ankylosing spondylitis patients compared to control
Das et al. (2019) [17]	181	Older adults	13:17	Several microbes	Osteoporosis	Different gut microbiota profiles were found to be associated with osteopenic, osteoporotic, and normal bone mass density
Fritzell et al. (2019) [25]	60	40 adults with lumbar disc herniation/lower back pain, 20 control patients with scoliosis	30:30	*Cutibacterium acnes*	Degenerative disc	*C. acnes* found in discs and vertebrae during surgery for disc herniation
Nilsson et al. (2018) [18]	90	Elderly women between the ages of 75 and 80 who have diminished bone mineral density	0:90	*Lactobacillus* *reuteri*	Osteoporosis	*L. reuteri* reduces total bone mass density compared to the placebo
Rajasekaran et al. (2017) [26]	22	15-disc herniations, 5- degenerate, 2-normal in MRI	15:7	*Propionibacterium acnes*	Proteome in intervertebral discs	Specific bacterial and host defense proteins were present in intervertebral discs
Rajasekaran et al. (2019) [27]	6 control discs, 5 degenerated discs	Group A (young 2nd–4th decades), Group B (aging. 5th–7th decade), Group C (degenerative discs)	4:7	---	Degenerative Disc	Unique proteome signatures of bacteria in discs of young, aging, and degenerative discs
Rajasekaran et al. (2020) [28]	24	8- brain-dead but living organ donors had healthy MRI discs, 8 had herniated discs, 8-disc degeneration	15:9	Several microbes	Degenerative disc	Distinct microbiome profiles in patients with healthy disc, disc herniations, and degenerative disc
Rao et al. (2020) [29]	812	NA	NA	*Cutibacterium acnes*	Degenerative disc	The research did not reveal any distinction in actual infection rates between the groups with non-degenerative and degenerative discs
Rettedal et al. (2020) [30]	86	All postmenopausal women: 18 osteoporosis, 42 osteopenia, 26 healthy controls	0:86	Bacteroides	Osteoporosis	Bacteroides taxa were more abundant in both osteopenia and osteoporosis
Scher et al. (2013) [31]	114	Rheumatoid Arthritis	11:33	*Prevotella copri*	Rheumatoid Arthritis	*P. copri* in stool is correlated with new onset untreated. rheumatoid arthritis
Scher et al. (2016) [15]	58	Rheumatoid Arthritis, Sarcoidosis, Control	43:25	Pseudonocardia	Rheumatoid Arthritis, Sarcoidosis	The composition of gut microbiota in individuals with rheumatoid arthritis and sarcoidosis was significantly decreased and is less varied in comparison to individuals without health issues
Wang et al. (2017) [19]	18	6 adults with primary osteoporosis, 6 with primary osteopenia, and 6 normal controls	3:15	Firmicutes and Bacteroidetes	Osteoporosis	Osteoporosis individuals contained an increased proportion of Firmicutes phyla but decreased proportion of Bacteroidetes compared to the control
Xu et al. (2020) [20]	96	48 primary osteoporosis patients and 48 healthy	37:59	Faecalibacterium and dialister	Osteoporosis	Increase in abundance of Faecalibacterium and dialister in patients with primary Osteoporosis

**Table 2 genes-14-01937-t002:** Microbiome and musculoskeletal disorders in animal studies.

Author	Sample Size	Intervention	MSK Disorder	Result
Guss et al. (2019) [32]	6–7 mice per group, 2 groups	Oral antibiotics vs. untreated	Osteoporosis	The decrease in microbiota synthesized vitamin K from the antibiotics led to a decrease in bone matrix quality
Guss et al. (2019) [32]	10–11 mice group 2 groups	Toll-le mceptor-5 deficient mice	Osteoarthritis	Gut microbiome may influence cartilage pathology
Hemandez et al. (2019) [33]	82 (40 modified microbiome, 42 untreated)	A tibial implant made of titanium, along with the introduction of Staphylococcus aureus in the synovial space.	Periprosthetic joint infection	Gut microbiota may influence susceptibility to periprosthetic joint infection The composition of gut microbiota could impact one’s vulnerability to periprosthetic joint infection
Li et al. (2016) [34]	10 mice per group 2 groups	Control vs. Probiotics	Osteoporosis	The microbiota within the gut lumen and heightened gut permeability contribute to the initiation of inflammatory pathways that are essential in causing bone loss in mice lacking sex steroids
Sjogren et al. (2012) [35]	485 mice	Germ free mice vs conventionally raised mice	Osteoporosis	In mice, the gut microbiota manages bone density by decreasing the production of inflammatory cytokines in both bone and bone marrow.
Wang et al. (2021) [36]	12 mice per group, 4 groups	Control, Control + *L. paracasei* S16 probiotic, Lumbar Doc Herniation (LDH), LDH+ *L. paracasei* S16 probiotics	Lumbar Disc Herniation (Low Back Pain)	*L. paracasei* S16 has the potential to alleviate LDH symptoms through the reduction of inflammation, modifications in gut microbiota, and alterations in serum metabolite
Yan et al. (2016) [37]	6 mice	Control vs. Antibiotics	Osteoporosis	The gut microbiota negatively impacts bone health, likely through IGF-1 mediation, causing a net anabolic deficit

## Data Availability

Not applicable.

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
