# Peer review of "The Human Microbiome and Its Role in Musculoskeletal Disorders"

_genes, 2023, doi:10.3390/genes14101937_

Round 1
Reviewer 1 Report
General Comments
The manuscript is generally well written and provides a general overview of the role of the human microbiome (MB) in relation to musculoskeletal disorders (MSDs). MB was analysed in human and animal studies. In human studies, MB was searched in health and diseases, including osteoporosis, rheumatoid arthritis, osteoarthritis, intravertebral disc degradation, scoliosis and inflammatory conditions. Also, the role of MB following musculoskeletal development and age-related sarcopenia was analysed. This study gives a new perspective in this field for further research.
However, the design of this paper should be slightly improved before publishing.
In my opinion, it is obligatory better to explain the novelty of this manuscript in the Introduction, because a lot of previous research analysed the role of human MB in health and diseases (How this study differ from the previous reviews and who can use this information).
Specific Comments
Introduction
Page 2. Please try better to explain the novelty of this manuscript (see General Comments).
Author Response
Thank you for your comment. We have added an additional paragraph at the end of the Introduction section of the main manuscript, which reads as follows:
"Currently, little is known about the interplay between the human microbiome and MSDs. The current narrative review evaluates emerging data on the role of the human microbiome in relation to MSDs. To our knowledge, a singular and comprehensive review on the role of the microbiome upon MSDs has not, to date, been addressed. Risk factors between many musculoskeletal conditions tend to overlap, suggesting crosstalk of these mechanisms. This crosstalk provides cause and reason to believe that a review of the microbiome in relationship to MSDs in this depth and coverage is needed and would provide a starting point for discussion and future research. This review is targeted to clinicians (e.g. orthopaedists, rheumatologists, physiatrists, physical therapists, etc) and basic science researchers interested in the topic of microbiome and MSDs."
Reviewer 2 Report
The manuscript titled: The Human Microbiome and its Role in Musculoskeletal Disorders represents various aspects of the human microbiota in relation to MSDs.
1. There are data that describe the roles of probiotics, particularly L. casei or L. acidophilus, in the management of rheumatoid arthritis in clinical and preclinical studies. Authors scarcely describe the role of probiobiotix in RA. Please see the review Paul AK, Paul A, Jahan R, Jannat K, Bondhon TA, Hasan A, Nissapatorn V, Pereira ML, Wilairatana P, Rahmatullah M. Probiotics and Amelioration of Rheumatoid Arthritis: Significant Roles of Lactobacillus casei and Lactobacillus acidophilus. Microorganisms. 2021 May 16;9(5):1070. doi: 10.3390/microorganisms9051070.
2. What is the clinical aspect of your review – should probiotics be regularly supplemented, and for how long - a few weeks/ months/ years?
3. Should the dose of supplemented probiotics differ between discussed diseases?
4. Which probiotic type, e.g., L. casei or L. acidophilus, Bifidobacterium, is preferred in specific diseases?
5. Could you give some recommendations (suggestions) in which disease the probiotics should be used, what type, and for how long?
Author Response
We like to thank the Reviewer for their comments and suggestions.
Comment 1:- There are data that describe the roles of probiotics, particularly L. casei or L. acidophilus, in the management of rheumatoid arthritis in clinical and preclinical studies. Authors scarcely describe the role of probiobiotix in RA. Please see the review Paul AK, Paul A, Jahan R, Jannat K, Bondhon TA, Hasan A, Nissapatorn V, Pereira ML, Wilairatana P, Rahmatullah M. Probiotics and Amelioration of Rheumatoid Arthritis: Significant Roles of Lactobacillus casei and Lactobacillus acidophilus. Microorganisms. 2021 May 16;9(5):1070. doi: 10.3390/microorganisms9051070.
Response: We have added this sentiment in the concluding remarks of the Rheumatoid Arthritis section.
Comment 2 - Probiotics, recommendations and guidelines
Response 2: We fully agree with the Reviewer that probiotics are very important and of great interest to the community. However, we hesitate to discuss more on this topic than what we already have noted in the paper. For one, our article is a broad narrative review. It is not a systematic review whereby specific questions are addressed and outcomes potentially synthesized. The request by the Reviewer goes beyond the scope of this current article. We do not feel comfortable providing any recommendations/guidelines as this was not the intent of this article. In addition, this article is broad, touching upon a multitude of disorders and not just one to gather the attention of probiotics linked to specific conditions.